# Cancer Stem Cell and Aggressiveness Traits Are Promoted by Stable Endothelin-Converting Enzyme-1c in Glioblastoma Cells

**DOI:** 10.3390/cells12030506

**Published:** 2023-02-03

**Authors:** Ignacio Niechi, José I. Erices, Diego Carrillo-Beltrán, Atenea Uribe-Ojeda, Ángelo Torres, José Dellis Rocha, Daniel Uribe, María A. Toro, Karla Villalobos-Nova, Belén Gaete-Ramírez, Gabriel Mingo, Gareth I. Owen, Manuel Varas-Godoy, Lilian Jara, Francisco Aguayo, Verónica A. Burzio, Claudia Quezada-Monrás, Julio C. Tapia

**Affiliations:** 1Laboratorio Biología Tumoral, Instituto de Bioquímica y Microbiología, Facultad de Ciencias, Universidad Austral de Chile, Valdivia 5110566, Chile; 2Millennium Institute on Immunology and Immunotherapy, Facultad de Ciencias, Universidad Austral de Chile, Valdivia 5110566, Chile; 3Facultad de Medicina Veterinaria y Recursos Naturales, Universidad Santo Tomás, Talca 3473620, Chile; 4Programa de Biología Celular y Molecular, Instituto de Ciencias Biomédicas, Facultad de Medicina, Universidad de Chile, Santiago 8380000, Chile; 5Centro de Biología Celular y Biomedicina, Facultad de Medicina y Ciencia, Universidad San Sebastián, Santiago 7510602, Chile; 6Faculty of Biological Sciences, Pontificia Universidad Católica de Chile, Santiago 8331150, Chile; 7Advanced Center for Chronic Diseases, Santiago 8330034, Chile; 8Millennium Institute on Immunology and Immunotherapy, Santiago 8331150, Chile; 9Centro Científico y Tecnológico de Excelencia Ciencia & Vida, Santiago 7750000, Chile; 10Programa de Genética, Instituto de Ciencias Biomédicas, Facultad de Medicina, Universidad de Chile, Santiago 8380000, Chile; 11Programa de Virología, Instituto de Ciencias Biomédicas, Facultad de Medicina, Universidad de Chile, Santiag 8380000, Chile; 12Department of Biological Sciences, Faculty of Life Sciences, Universidad Andrés Bello, Santiago 8370146, Chile

**Keywords:** glioblastoma, stemness, endothelin, CK2, aggressiveness

## Abstract

Glioblastoma (GBM) is the most common and aggressive type of brain tumor due to its elevated recurrence following treatments. This is mainly mediated by a subpopulation of cells with stemness traits termed glioblastoma stem-like cells (GSCs), which are extremely resistant to anti-neoplastic drugs. Thus, an advancement in the understanding of the molecular processes underlying GSC occurrence should contribute significantly towards progress in reducing aggressiveness. High levels of endothelin-converting enzyme-1 (ECE1), key for endothelin-1 (ET-1) peptide activation, have been linked to the malignant progression of GBM. There are four known isoforms of ECE1 that activate ET-1, which only differ in their cytoplasmic N-terminal sequences. Isoform ECE1c is phosphorylated at Ser-18 and Ser-20 by protein kinase CK2, which increases its stability and hence promotes aggressiveness traits in colon cancer cells. In order to study whether ECE1c exerts a malignant effect in GBM, we designed an ECE1c mutant by switching a putative ubiquitination lysine proximal to the phospho-serines Lys-6-to-Arg (i.e., K6R). This ECE1c^K6R^ mutant was stably expressed in U87MG, T98G, and U251 GBM cells, and their behavior was compared to either mock or wild-type ECE1c-expressing clone cells. ECE1c^K6R^ behaved as a highly stable protein in all cell lines, and its expression promoted self-renewal and the enrichment of a stem-like population characterized by enhanced neurospheroid formation, as well as increased expression of stem-like surface markers. These ECE1c^K6R^-derived GSC-like cells also displayed enhanced resistance to the GBM-related chemotherapy drugs temozolomide and gemcitabine and increased expression of the ABCG2 efflux pump. In addition, ECE1c^K6R^ cells displayed enhanced metastasis-associated traits, such as the modulation of adhesion and the enhancement of cell migration and invasion. In conclusion, the acquisition of a GSC-like phenotype, together with heightened chemoresistance and invasiveness traits, allows us to suggest phospho-ECE1c as a novel marker for poor prognosis as well as a potential therapeutic target for GBM.

## 1. Introduction

High grade IV glioma, also termed glioblastoma (GBM), is the most common and aggressive brain tumor, displaying a high proliferation rate, cellular heterogeneity, and invasive potential [1,2,3]. GBM accounts for approximately 70% of all gliomas, and diagnosed patients display an overall survival rate of only 26% after one year [3,4]. Despite multimodal treatments, which include tumor resection followed by radiotherapy in combination with temozolomide (TMZ) or gemcitabine (GEM) chemotherapy [2,5,6,7], there has been no improvement in patient survival due to early tumor recurrence [2,8,9]. Recurrence is essentially mediated by a small subpopulation with stem cell characteristics within the tumor, named glioblastoma stem-like cells (GSCs), which mediate resistance to neoplastic drugs and infiltrate healthy tissues [10,11,12,13]. Chemoresistance in GSCs is mainly due to the overexpression and activation of efflux transporters that pump drugs out of the cells, such as ABCC1, ABCG2, and ABCC3, among others [5,14]. However, GSCs have also been shown to activate signaling pathways that promote the expression of epithelial–mesenchymal transition (EMT) markers and matrix metalloproteases (MMPs), which are associated with their invasive phenotype [15,16]. Thus, GBM recurrence is a result of stemness promotion in the tumor niche, which generates aggressiveness traits such as chemoresistance and infiltration. These are the most important traits to beat in the development of therapeutic strategies against GBM [6,11].

Several factors in the tumor niche have been related to GBM aggressiveness and stemness phenotypes, including adenosine [11], VEGF [17], and endothelin-1 [18]. Endothelin-1 (ET-1) is a small mitogenic peptide activated by endothelin-converting enzyme-1 (ECE1) on the ET-1 axis [19]. Recently, ECE1 has emerged as a potential target since it promotes aggressiveness in breast, ovarian, prostate, and colorectal cancer cells [20]. ECE1 is expressed as four isoforms, depending on the cell type, which are differentiated at their cytoplasmic N-termini [20,21,22]. Interestingly, the N-terminus of the ECE1c isoform is phosphorylated at Ser-18 and Ser-20 by protein kinase CK2, boosting its stability and enhancing the invasiveness of colorectal cancer cells, albeit by an unknown mechanism in which a putative ubiquitination site, lysine 6, could have a potential role [23,24]. CK2 was suggested as a metastasis-associated gene in a proteomic study on different cancer cell lines [25], and its levels correlate with poor patient prognosis [26]. CK2 is elevated in a wide variety of cancers, including GBM [27,28], which has been associated with increased growth and proliferation, in addition to GSC maintenance [29,30]. We recently reported that the inhibition of CK2 with silmitasertib promotes early methuosis-like cell death and thereby decreases tumorigenicity for longer times in colorectal cancer cells [31]. CK2 is thought to promote cell survival through the phosphorylation of β-catenin through the canonical Wnt pathway [30], which increases the expression of many transcriptional targets, including survivin and cyclooxygenase-2 [32,33,34]. Indeed, ET-1 is another β-catenin target, and its expression is increased in several cancers, including GBM [19]. Moreover, ECE1 expression is known to be increased in GBM tumors in comparison to healthy tissue, and its pharmacological inhibition decreases in vitro proliferation [35], but whether ECE1c plays a role in GBM stemness is unknown. Here, we show that the expression of a super-stable ECE1c mutant conduces to a GSC-like phenotype in GBM cell lines, as evaluated by stemness, chemoresistance, and invasiveness traits, which leads us to suggest phospho-ECE1c as a novel marker for poor prognosis and thereby a potential target for GBM therapy.

## 2. Materials and Methods

### 2.1. Cell Culture

All cell lines were purchased from ATCC (Manassas, VA, USA). U87MG cells were purchased by Dr. Quezada-Monrás, while T98G and U251 cells were purchased and gently donated by Dr. Varas-Godoy. Once they arrived at the laboratory, the cells were immediately expanded in DMEM-F12 medium supplemented with 10% FBS, 100 U/mL penicillin, and 100 μg/mL streptomycin (Gibco, Waltham, MA, USA) at 37 °C and 5% CO_2_, followed by storage in liquid nitrogen at −190 °C. Once a year, one N_2_ aliquot was thawed, expanded, and stored again at −80 °C. For experiments, one −80 °C aliquot was thawed and grown in normal media. All experiments were performed within one year, and cells were eliminated after 15 passages, as requested by each local biosecurity committee. Mycoplasma contamination was tested monthly using the EZ-PCR Mycoplasma Test kit (Biological Industries, Beit Haemek, Israel), with the last test being performed six months ago and yielding no contamination.

### 2.2. GlioVis Analysis

Gene expression and survival data for patients diagnosed with GBM were obtained from records collected by the Cancer Genome Atlas (TCGA). They were analyzed using the GlioVis web service (Version 0.20) (http://gliovis.bioinfo.cnio.es/, accessed on 10 May 2021) [36]. Only adult patients whose tumors had mRNA expression obtained by RNA-seq for our genes of interest were included. Pairwise comparisons were performed using a t-test (with a Bonferroni correction). The *p*-values of the pairwise comparisons are indicated on the graphs as *** *p* < 0.001.

### 2.3. Lentiviral Cloning 

Full-length wild-type ECE1c cDNA with an in-frame 5′-Flag-tag was previously cloned by us in the bicistronic lentiviral plasmid pLVX-IRES-mCherry (Clontech, Mountain View, CA, USA). Lys-6-Arg-site-directed mutagenesis was performed using the GENEART kit (Thermo Fisher, Waltham, MA, USA) according to manufacturer’s instructions. Lentiviral vector production was carried out using the Lenti-X™ 293T cell line (Clontech) by transfecting a second-generation lentiviral system using a calcium phosphate protocol [37]. Briefly, Lenti-X™ 293T cells were transfected with 8 g of psPax2 (encoding the Gag-Pol protein), 4 g of pCMV-VSVg (encoding the VSV G-glycoprotein), and 8 μg of pLVX-IRES-mCherry (encoding ECE1c^WT^, ECE1c^K6R^, or an empty control, termed “mock”) and then suspended in 500 μL of 250 mM CaCl_2_. At 48 h post-transfection, supernatants containing pseudotyped particles were harvested and passed through a cellulose acetate filter with a pore size of 0.45 μm. Viral particles were purified and concentrated by ultracentrifugation at 28,000 rpm for 75 min in a SureSpin 630 rotor (Thermo Fisher, Waltham, MA, USA) through a 25% sucrose cushion (TNE-Sucrose 25%). U87MG, T98G, and U251 cells were cultured at a density of 5 × 10^4^ cells/well in 12-well culture plates, along with recombinant lentiviruses at an MOI of 5 under normal growth conditions. The expression of mCherry was examined 72 h post-infection under a Nikon Eclipse TS100 inverted epifluorescence microscope. Cells were expanded for 1–2 weeks, trypsinized, and sorted using a FACSAria Fusion instrument (Becton Dickinson, Franklin Lakes, NJ, USA). Gating was performed on the brightest mCherry cells, which were collected, expanded for a further 1–2 weeks, and subsequently sorted for a second time to obtain >99% pure clones.

### 2.4. Glioblastoma Stem-like Cell (GSC) Enrichment

U87-MG cells were grown at 37 °C in neurobasal medium (Gibco) supplemented with 20 ng/mL EGF (PeproTech, Cranbury, NJ, USA), 20 ng/mL bFGF (PeproTech), 1X B27 (w/o vitamin A, Gibco), 100 U/mL penicillin, 100 U/mL streptomycin (Gibco), and 2 mM L-glutamine (Gibco), as described elsewhere [13]. The medium was replaced every 2 days.

### 2.5. Cell Viability 

Cells (5 × 10^3^ cells/well) were plated into 96-well plates, grown overnight, and then treated with DMSO, 400 μM temozolomide (TMZ, Santa Cruz, Santa Cruz, CA, USA), and 4 μM gemcitabine (GEM, Fresenius Kabi, Hamburg, Germany) for 24 h. Cell viability was determined using an MTS assay (Promega, Madison, WI, USA), according to manufacturer’s instructions.

### 2.6. Cell Adhesion

A cell adhesion assay was performed as described elsewhere [13]. Briefly, cells (2.5 × 10^4^ cells/well) were plated into 96-well plates that were precoated with 2 μg/mL fibronectin and incubated for 5–90 min at 37 °C. Debris and dead cells were carefully removed, and cells were washed with PBS. Cells were then fixed in 3.7% paraformaldehyde (PFA) for 10 min and stained for 10 min in a crystal violet solution (1% crystal violet in 20% methanol). Cells were washed with PBS, and the crystal violet staining was released with 10% acetic acid. Cell adhesion was determined by measuring the optical density at 540 nm using a microplate reader.

### 2.7. Indirect Immunofluorescence (IIF)

IIF was performed as described elsewhere [13]. Briefly, spheres were formed as in Section 2.4, seeded onto pretreated poly-lysine coverslips, and incubated overnight with primary antibodies against CD44 (Cell Signaling 3570S, Danvers, MA, USA) and Nestin (Abcam 22035, Waltham, MA, USA). After washing, the samples were incubated with Alexa-488 and 300 nM DAPI (Thermofisher) for 1 h. The samples were fixed with DAKO (Agilent Technologies, Santa Clara, CA, USA) and visualized using epifluorescence microscopy (Zeiss, Oberkochen, Germany).

### 2.8. Migration and Invasion

Migration and invasion assays were performed as described elsewhere [13]. Cells (7.5 × 10^4^ cells/chamber) were plated on the upper side of a polycarbonate Transwell chamber (6.5 mm, 8.0 μm, Corning, Corning, NY, USA) for the migration assay or in a Matrigel-coated Transwell chamber (Corning) for the invasion assay. In both cases, cells were seeded in serum-free DMEM-F12. As a chemoattractant, the bottom chamber contained DMEM-F12 supplemented with 10% FBS. Cells were incubated at 37 °C for 2 h (migration) or 16 h (invasion). Cells in the top chamber were carefully removed with cotton swabs, and cells that crossed through the chamber were fixed in 3.7% PFA and stained in a crystal violet solution for 10 min. Cells were counted using the 10× objective in 5 different fields of the underside of the insert. The mean number of cells was normalized to 1 using the mock conditions and then plotted.

### 2.9. Vasculogenic Mimicry

HEY-A8 (ovarian adenocarcinoma) cells were used to determine the optimal number of cells needed to obtain vascular structures, as described elsewhere [38]. Experiments were performed with 70–80% confluent HEY-A8 cells incubated with media from ECE1c^WT^- or ECE1c^K6R^-expressing T98 and U251 cells to study whether these conditioned media may alter the capacity of HEY-A8 cells to undergo vasculogenic mimicry in cell cultures. In brief, 12 × 12 mm glass coverslips (Marienfeld, Lauda-Königshofen, Germany) were washed in ethanol, air-dried, placed in 12-well culture plates coated with 12.5 μL of Matrigel (Corning) per coverslip, and air-dried for 45–60 min at 37 °C in an incubator. Cell cultures were trypsinized and counted, and 15,000 cells were resuspended in 50 μL of culture medium that was seeded on Matrigel-coated coverslips. Cells were incubated at 37 °C for 1 h to allow their adhesion to the matrix and then covered with 3 mL of culture medium (RPMI 1640 supplemented with 15% FBS) or the conditioned medium taken from T98G and U251 cells after 24 h of culture. The formation of tubular structures was followed over a 4-day period.

### 2.10. Protein Stability

Protein stability assays were performed as described elsewhere [23,24]. Briefly, cells (1 × 10^6^ cells/well) were seeded into 12-well plates, cultured overnight at 37 °C and 5% CO_2_ in DMEM-F12 medium supplemented with 10% FBS, and incubated overnight with 20 μg/mL cycloheximide (CHX, Tocris, Bristol, Avon, UK) to inhibit protein synthesis. Then, cells were incubated in the presence of 25 μM silmitasertib (a selective CK2 inhibitor, formerly known as CX-4945; Apexbio, Houston, TX, USA) or vehicle only (0.001% DMSO) for 6 h. Cells were then harvested and lysed, and the total protein was analyzed using Western blots.

### 2.11. RT-PCR

Total RNA was extracted with Trizol (Gibco) and quantified using a NanoDrop device. Reverse transcription was performed on 1 μg of total RNA with MMLV-RT (Thermo-Fisher), following the manufacturer’s instructions. Quantitative PCR (qPCR) was performed using the 2^−ΔΔCt^ method, with GAPDH as a normalizer gene (14) with 250 nM of each primer using the 5x HOT FIREPol EvaGreen qPCR Mix Plus (ROX) (Solis BioDyne, Tartu, Estonia), following manufacturer’s instructions. The primers that were used included CD133, F: 5′-CCAGCTGAATAGCAACCCTGAACT-3′, R: 5′-ACCAGGCCATCCAAATCTGTCCTA-3′; CD44, F: 5′-AGGACAGAAAGCCAAGTGGACTCA-3′, R: 5′-CGACTCCTTGTTCACCAAATGCAC-3′; ABCC1, F: 5′-AGGTCAAGCTTTCCGTGTAC-3′, R: 5′-GGACTTTCGTGTGCTCCTGA-3′; ABCG2, F: 5′-TTCGGCTTGCAACAACTATG-3′, R: 5′-TCCAGACACACCACGGATAA-3′; ABCC3, F: 5′-CCTTTCTGTGTCCTACTCCTTG-3′, R: 5′-CGCCTCTGTCTCTGTCTTG-3′; SNAIL, F: 5′-TTCTCACTGCCATGGAAT-3′, R: 5′-GCAGAGGACACAGAACCAG-3′; TWIST, F: 5′-GGCCGGAGACCTAGATGTC-3′, R: 5´-CCACGCCCTGTTTCTTTGAAT-3′; CDH1, F: 5′-GAGGAATCCAAAGCCTCAGGTCAT-3′, R: 5′-TCACCCACCTCTAAGGCCATCTTT-3′; GFAP, F: 5′-TCGATCAACTCACCGCCAACA-3′, R: 5′-CCAGGGTGGCTTCATCTGCTT-3′; NES, F: 5′-AGAGGGCAAAGTGGTAAGCA-3′, R: 5′-AGTGTCTCATGGCTCTGGTT-3′; EDN1, F: 5′-TGGGAAAAAGTGTATTTATCAGCA-3′, R: 5′-TTTGACGCTGTTTCTCATGG-3′.

### 2.12. Western Blot 

Proteins (30–40 μg) were separated using SDS-PAGE, transferred to nitrocellulose membranes, and stained with Ponceau Red. According to the MW markers and Ponceau patterns, membranes were cropped to detect different proteins and were separately blocked with blocking buffer (PBS/0.05% Tween/5% non-fat milk). Membranes were incubated at 4 °C overnight with primary antibodies, followed by incubation for 1 h with a secondary HRP-conjugated anti-IgG antibody (Jackson). The primary antibodies were Flag (Cell Signaling D6W5B), Snail (Abcam 135708), Twist (Abcam 49254), E-cadherin (Abcam 76055), β-actin (Santa Cruz Biotechnology 47778), MMP9 (Santa Cruz Biotechnology 4778), Nestin (Abcam 22035), CD44 (Cell Signaling 3570S), GFAP (Cell Signaling 3670S), TUBB3 (Sigma Aldrich 4700544), and Sox2 (Cell Signaling 3579S). Bands were revealed using the West Dura chemiluminescence system (Thermo-Fisher), and imaging was performed on a Syngene G:Box instrument (Synoptics, Cambridge, UK).

### 2.13. Enzyme-Linked Immunosorbent Assay (ELISA)

Endothelin-1 was quantified in culture media using an Endothelin-1 (ET-1) Human ELISA Kit (Invitrogen, EIAET1). Cells at a density of 10^4^ cells/well were incubated for 48 h, as described in Section 2.1. The ET-1 levels (pg) were measured according to the manufacturer’s instructions and normalized to 1 mg/mL of total protein [24].

### 2.14. Statistical Analysis

A statistical analysis was performed and graphics were created using GraphPad Prism 6.01 software. Values were plotted as means ± SDs from at least three independent experiments. A statistical analysis was performed on raw data using the Peritz F multiple means comparison test. Student’s *t*-test was used for unpaired data. *p* ≤ 0.05 was considered statistically significant.

## 3. Result

### 3.1. ECE1 Transcript Expression Is Related to Aggressive GBM Subtypes

An in silico transcriptomic analysis using public human RNA-seq datasets was performed. This analysis showed that ECE1 mRNA levels were increased in glioblastoma (GBM) tumors compared to lower-grade gliomas, including oligodendroglioma, oligoastrocytoma, and astrocytoma (Figure 1A). In addition, ECE1 mRNA levels were higher in GBM tumors classified with classical and mesenchymal phenotypes, whose invasive potentials were higher in comparison to tumors with a proneural phenotype (Figure 1B). Despite the human datasets that were used not providing any information on specific isoforms, this transcriptomic analysis suggests that augmented ECE1 expression may have a role in GBM progression and aggressiveness.

### 3.2. Lysine 6 Is Involved in CK2-Dependent ECE1c Stability

The primary sequence of ECE1c from several species shows a conserved lysine at position 6 and is close to serines 18 and 20, which are involved in ECE1c stability by CK2-dependent phosphorylation [20,23]. In order to assess the role of this residue in ECE1c stability in GBM cells, a Lys-6-to-Arg (K6R) point mutation was created using lentiviral vectors to express a flag-tagged super-stable ECE1c^K6R^ or wild-type ECE1cWT in U87MG, T98G, and U251 glioblastoma cell lines. Cells were incubated with 20 μg/mL cycloheximide (CHX) in order to inhibit protein synthesis. Flag-tagged ECE1c levels were detected using Western blots for up to 6 h. The results showed that ECE1c^K6R^ was extremely stable in comparison to ECE1cWT in all GBM cell lines (Figure 2). Particularly in U87MG cells, the ECE1c^WT^ levels dropped to 50% at 3 h in the absence of the specific CK2 inhibitor silmitasertib (25 μM). However, in the presence of the inhibitor, the levels decreased to around 15%, confirming the role of CK2-mediated phosphorylation in the stability of ECE1c. In contrast, the ECE1c^K6R^ mutant displayed enhanced stability over ECE1c^WT^, both in the presence and absence of silmitasertib, only dropping to around 60% at 6 h in both cases (Appendix A). Altogether, these data suggest that lysine 6 is involved in CK2-dependent ECE1c stability.

### 3.3. Highly Stable ECE1c^K6R^ Promotes Stem-like Traits in GBM Cells

Given the key role of GSCs in the development and aggressiveness of GBM, we evaluated mock, ECE1c^WT^-expressing, and ECE1c^K6R^-expressing cell clones on their ability to modulate stemness traits. As CD44 and CD133 markers have been related to a stem-like phenotype and used for GSC detection [38,39,40], mRNA levels were quantified using RT-qPCR in U87MG cell clones. ECE1c^K6R^ expression led to significantly higher levels of both markers compared to mock and ECE1c^WT^-expressing cells (Figure 3). These changes also paralleled qualitative changes of some markers determined using conventional RT-PCR in T98G and U251 cell clones (Appendix A) and the IIF of CD44 and Nestin as stemness markers (Appendix A).

To study a specific effect of either ECE1c^WT^ or ECE1c^K6R^ expression on neurosphere formation, U87MG clones were grown in neurobasal medium for only 5 days before saturation (Figure 4A). As observed, more than twice as many spheres were obtained in ECE1c^K6R^ cells compared to mock and ECE1c^WT^ cells (Figure 4B), while spheres from either ECE1c^K6R^ or ECE1c^WT^ cells were three times larger than those from mock cells (Figure 4C). Altogether, these results suggest that the expression of the super-stable ECE1c^K6R^ protein in GBM cells leads to sphere formation under anchorage-independent conditions.

### 3.4. Chemoresistance Is Enhanced in ECE1cK6R-Expressing GBM Cells

The viability of the three different cell clones was measured under normal culture conditions using MTS in U87MG cells. As observed, the viability of ECE1c^K6R^-expressing cells was only around 10–20% higher in comparison to ECE1c^WT^ and mock cells, respectively (Figure 5A). The GSC phenotype is related to acquisition of chemoresistance [10,11]. Thus, viability in the presence of traditional GBM antineoplastic drugs was evaluated in the three cell clones. As already reported for these highly chemoresistant U87MG cells [10,14], the viability of mock cells decreased by 20% in treatments with 400 μM TMZ or 4 μM GEM for 24 h. However, ECE1c^K6R^ and ECE1c^WT^ cells maintained their viability in the presence of both drugs (Figure 5B). This behavior could be a consequence of the enhanced expression of the ABCG2 efflux pump in both ECE1c-expressing cell clones, which was somewhat higher, albeit not significantly, in ECE1c^K6R^ compared to ECE1c^WT^ (Figure 5C). These results suggest that the overexpression of ECE1c promotes a chemoresistant phenotype in GBM cells.

### 3.5. ECE1cK6R Promotes Metastasis-Associated Traits in GBM Cells

To evaluate the migration and invasion potential of GSCs, we evaluated the effect of the overexpression of ECE1c^WT^ or ECE1c^K6R^ on U87MG cell adhesion. No significant differences were observed between mock and ECE1c^WT^ cells. However, adhesion was significantly higher in ECE1c^K6R^ cells (Appendix A). This behavior was replicated in a 3D migration assay since the ECE1c^K6R^ mutant promoted enhanced migration efficiency through the transwell membrane in all cell clones (Figure 6A–C). The migration capacity enhanced by ECE1c was accompanied by changes in EMT marker expression, such as Twist, Snail, and E-cadherin. The mRNA (Figure 7A) and protein (Figure 7B,C) levels were significantly altered following ECE1c^K6R^ overexpression in GBM cells. Likewise, as described for several cancer cell lines [20], the invasion capability of ECE1c^WT^-expressing cells was either higher than or equal to mock cells, depending on the cell type. Nevertheless, the expression of the super-stable ECE1c^K6R^ mutant significantly enhanced the invasiveness of all three GBM cell lines (Figure 8).

Finally, considering that the catalytic product of ECE1c, the endothelin-1 peptide (ET1), plays a role in vascularization and exogenously supplied ET1 induces migration and MMP expression in U251 cells [13,41], we wished to determine if ET1 expression was induced and, if so, to determine the potential effect of this induction on vasculogenic mimicry (VM) in our GBM cell clones. First, extracellular levels of ET1 were measured using ELISA. As expected, we observed increases in ET1 levels in both ECE1c^WT^- and ECE1c^K6R^-expressing cells, which were similar after 48 h (Appendix A). This result indicated that the super-stability of ECE1c^K6R^ did not affect the catalytic production of ET1, which was indistinguishable from that of ECE1c^WT^-expressing cells. Notably, the presence of ET1 in conditioned media taken from either ECE1c^WT^- or ECE1c^K6R^-expressing cells did not alter the capacity of HEY-A8 cells to undergo VM in comparison to both standard and mock cell media (Appendix A–I). Altogether, these results suggest that the secreted levels of ET1 are not enough to explain the significant increases in aggressiveness traits, including stemness gene expression, self-renewal capacity, drug resistance, tumorigenesis, and invasiveness potential, observed in vitro in the super-stable ECE1c^K6R^-expressing GBM cells (Figure 9).

## 4. Discussion

Most proteins are tagged by lysine ubiquitination for degradation [39,40,41,42]. The N-terminus domain of ECE1c contains a Lys-6 in proximity to the Ser-18 and Ser-20 residues, which are phosphorylated by CK2, conferring stability to the protein [20,24]. In a search for the occurrence of natural mutations at Lys-6, which could shed light on a role of ECE1c in GBM patients, 585 samples were analyzed in silico with the cBioPortal software accessed on 18 April 2021 (https://www.cbioportal.org). This genomic study showed only two missense mutations, L202F and G132D, which are in the catalytic domain of ECE1c. Thus, the ECE1c^K6R^ mutation was not found in this population of GBM patients. However, CK2 has indeed been found to be aberrantly increased in many cancers, including GBM [25,26,27,28]. Moreover, we have reported that Ser-18 and Ser-20 are phosphorylated by CK2 in ECE1c, increasing its stability and enhancing the aggressiveness of colon cancer cells [23]. This led us to hypothesize that a putative ubiquitination site, Lys-6, could have a potential role in the gain of ECE1c stability and the enhanced aggressiveness in GBM. Hence, although the ECE1c levels are unaltered in GBM tumors, aberrantly increased CK2 may promote its stability through phosphorylation, enhancing tumor aggressiveness and relapse potential. Thus, ECE1c may contribute to aggressiveness in a CK2-dependent manner (Figure 9) [20]. 

Here, we have shown that a Lys-6-to-Arg mutant enzyme, ECE1c^K6R^, is highly stable in comparison to its normal counterpart, ECE1c^WT^, in three different GBM cell lines, namely U87MG, T98G, and U251 cells. Of note, the effects seem to be phosphorylation-dependent since the pharmacological inhibition of CK2 with silmitasertib mainly decreased the ECE1c^WT^ protein levels, which indicated a key role of Lys-6 in its stability. Our findings suggest a post-translational mechanism for enhancing ECE1c protein stability, which would involve the CK2-dependent phosphorylation of Ser-18 and Ser-20 [23]. Although we did not address the issue of how Lys-6 is not ubiquitinated upon phosphorylation by CK2, our reported data indicate that ECE1c stability is indeed linked to phosphorylation and consequently proteasome degradation since the levels of a recombinant protein of the N-terminal end of ECE1c fused to GFP are restored when the proteasome inhibitor MG-132 is used in the presence of a CK2 inhibitor [21]. It is notable that ubiquitination does not occur at arginine or another residue. Therefore, our results and those with MG-132 strongly suggest that the super-stability of the ECE1c^K6R^ mutant is indeed a consequence of ubiquitination blockage upon the switch of lysine to arginine. 

In addition, the literature indicates that CK2 may regulate the stability of several proteins; for example, OTUB1 deubiquitinase phosphorylation promotes its nuclear activity, deubiquitinating and stabilizing chromatin binding proteins [43,44,45]. Moreover, the CK2 phosphorylation of c-Myc prevents its proteasomal degradation, enhancing the transcription of genes involved in various hallmarks of cancer [46]. The simplest mechanism for explaining the enhanced stability of ECE1c^K6R^ is a putative conformational change of the phosphorylated N-terminus domain of ECE1c, which may prevent its ubiquitination. Another plausible mechanism is the binding of an adaptor to the phospho-sites in a manner similar to EGFR signaling. Indeed, this has been suggested for HSP90, which promotes the expression of MDR1 due to resistance to paclitaxel in colorectal cancer cells [47].

The display of stem-like cell traits of ECE1c^K6R^-expressing GBM cells is a novel finding of this work, which strongly suggests a link between the endothelin-1 (ET1) axis and the occurrence of GSCs. Indeed, an increase in endothelin A receptor (ETAR) expression has been observed in CD133+ ovarian cancer stem-like cells and primary cultures, and sphere formation was reduced by using receptor antagonists in combination with chemotherapy [48]. In this work, we observed significant differences in neurosphere formation between ECE1c^WT^- and ECE1c^K6R^-expressing cells after only 5 days of culture in a supplemented neurobasal medium. As we reported previously [13,14,17], neurosphere formation was similar both in size and number after 7 days of growth, which indicates a saturating effect. However, dramatically elevated levels of CD133 and CD44 were only detected in ECE1c^K6R^ cells, which agrees with an upregulating effect of this protein on a GSC phenotype.

Our findings show that ECE1c promotes chemoresistance in GBM cells. Although the difference in viability of around 10% does not explain the enhanced resistance of ECE1c^K6R^ cells to TMZ or GEM, this observation is consistent with the fact that ECE1 inhibition decreases proliferation, as already described for GBM cells [35]. In relation to the above, the fact that only ABCG2 expression was increased in ECE1c-overexpressing cells is consistent with results already reported for U87MG cells, in which chemoresistance to TMZ, and possibly also GEM, is mediated by an ABCG2 pump [49]. Moreover, it seems that increased ABCG2 expression is not sufficient to create a difference between ECE1c^K6R^ and ECE1c^WT^ cells, as ABCC1 and ABCC3 are involved in resistance to other drugs such as vincristine or tacrolimus [10,11,14,50]. However, convincing evidence links the ET1 axis with chemoresistance both in GBM and other cancers [51]. For example, antagonism of both ET1 receptors with macitentan led to sensitization to TMZ as well as long-term survival in an orthotopic murine model of GBM [52].

The increased migratory and invasive potentials of ECE1c^K6R^-expressing GBM cells were probably consequences of the differences in the expression of EMT markers. This may be similar to ovarian cancer cells, in which ECE1c levels are correlated with invasiveness, EMT, and ET1 levels [53]. This suggests that in GBM cells the effect of ECE1c may also be dependent on the production of ET1. Indeed, exogenous ET1 induces migration and MMP expression in U251 GBM cells [42], with a known role of MMP-9 in GBM invasiveness [13]. However, exogenous ET1 does not rescue the negative effects of ECE1 silencing on the invasiveness of prostate cancer cells [54], suggesting an effect that is independent of ET1 production. In addition, ET1 production in this work was indistinguishably higher in both ECE1c^K6R^- and ECE1c^WT-^expressing GBM cells, although the presence of ET1 in their growth media did not alter the capacity of HEY-A8 cells to undergo VM, suggesting that secreted ET1 is not enough to explain the enhanced aggressiveness observed in our GBM cell clones. Nevertheless, the ET1 levels were measured after 48 h of growth, which may be too late to observe significant differences between ECE1c^K6R^ and ECE1c^WT^ clones. Therefore, differential ET1 levels during earlier stages cannot be ruled out, with a potentially significant impact on the stemness traits observed during later stages of growth.

We have shown here that the mutation of Lys-6-to-Arg significantly boosts ECE1c stability, and surprisingly its overexpression dramatically enhances in vitro traits attributable to GSCs in GBM cells. As stated above, we did not encounter data concerning a natural occurrence of the Lys-6 mutation in GBM patients. However, our in vitro results provide a proof-of-concept study linking the stability of ECE1c with the aggressiveness of GBM in several cell lines. The above suggests a post-translational mechanism for enhancing the aggressiveness of GBM cells, which would involve the CK2-dependent phosphorylation of ECE1c. Although we did not address the issue of how Lys-6 is not ubiquitinated upon the phosphorylation of Ser-18 and Ser-20 by CK2, our reported data indicate that ECE1c stability is indeed linked to phosphorylation and consequently proteasome degradation since the levels of a recombinant protein of the N-terminal end of ECE1c fused to GFP were restored when the proteasome inhibitor MG-132 was used in the presence of a CK2 inhibitor [21]. Thus, a super-stable ECE1c leading to an aggressive GSC phenotype in GBM cells is an issue that may be occurring in patients. We think our data uncovered the existence of a post-translational mechanism enhancing the aggressiveness of GBM cells, which involves the CK2-dependent phosphorylation of ECE1c, as no mutations at K6, S18, or S20 have been found (personal communication). Our findings suggest that in GBM ECE1c (augmented according to TCGA) may be phosphorylated at S18 and S20 by CK2 (also augmented, as shown at [27,28]), promoting stemness traits and making the tumor more aggressive, leading to recurrence and/or metastasis and therefore poor patient prognosis. Thus, ECE1c could contribute to cancer aggressiveness in a post-translational manner, as we have suggested elsewhere [20]. However, although the mechanism of how CK2-mediated phosphorylation enhances the stability of ECE1c is currently unclear, to our knowledge this is the first time that the super-stability of ECE1c has been associated with the stem-like and aggressiveness traits of GBM cells. Therefore, our findings suggest a novel role of phospho-ECE1c as a marker of poor prognosis and a potential target for the treatment of this disease.

## Figures and Tables

**Figure 1 cells-12-00506-f001:**
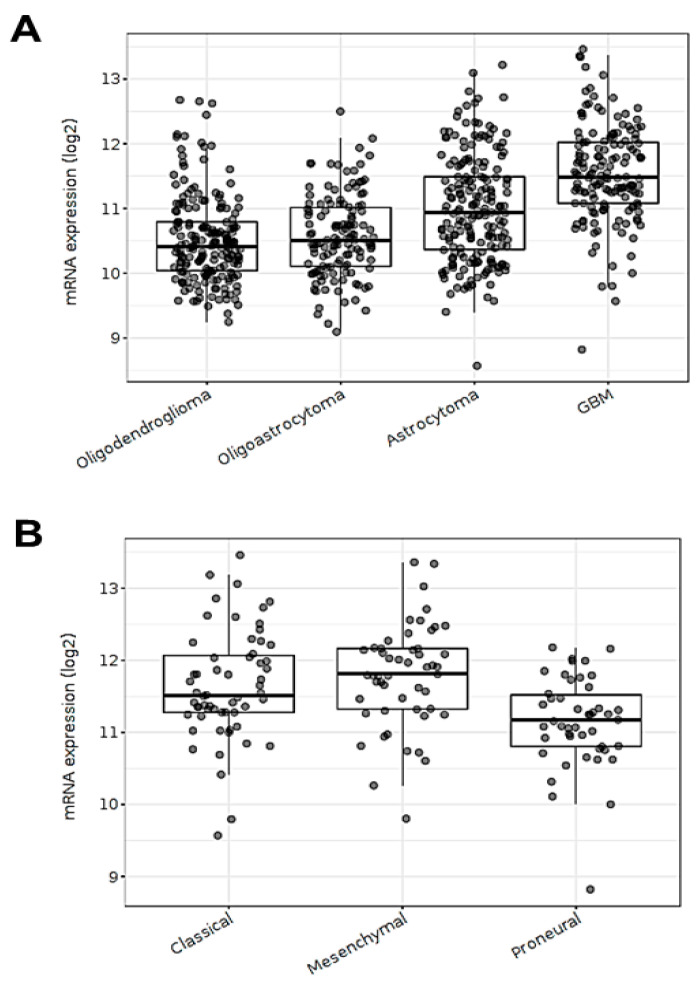
ECE1 mRNA levels are associated with highly aggressive gliomas. (**A**) ECE1 mRNA levels according to glioma grade (oligodendroglioma (*n* = 191), oligoastrocytoma (*n* = 130), and astrocytoma (*n* = 194)) and GBM (*n* = 152). (**B**) ECE1 mRNA levels according to subtypes of GBM. The samples were classified as classical (*n* = 59), mesenchymal (*n* = 51), and proneural (*n* = 46) subtypes. Glioma samples were obtained from The Cancer Genome Atlas (TCGA) RNA-seq database and analyzed using GlioVis accessed on 18 April 2021 (http://gliovis.bioinfo.cnio.es/) [36]. In total, 667 glioma samples were selected to analyze the differential ECE1 mRNA levels of low-grade gliomas and GBM. The GBM samples were selected to analyze the differential ECE1 mRNA levels of the GBM subtypes.

**Figure 2 cells-12-00506-f002:**
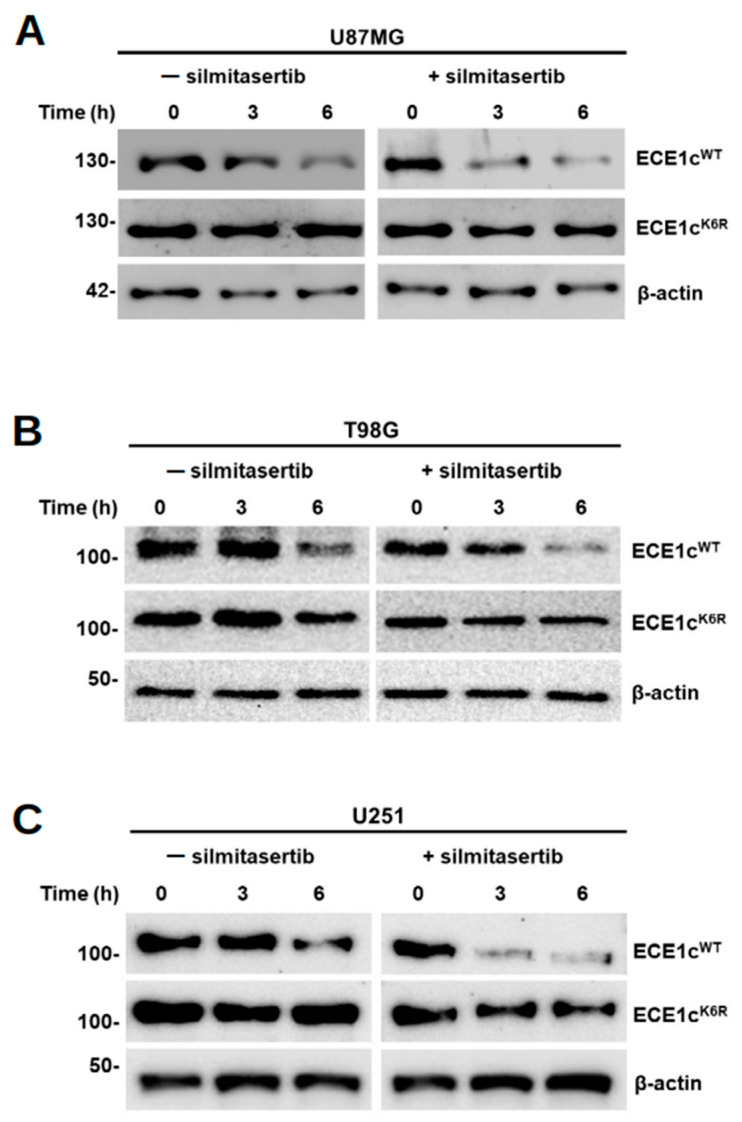
ECE1c^K6R^ mutant was highly stable in GBM cells. Flag-tagged ECE1c^WT^- or ECE1c^K6R^-expressing U87MG (**A**), T98G (**B**), and U251 (**C**) cells were treated with 20 μg/mL cycloheximide (CHX) in the absence or presence of 25 μM silmitasertib for 6 h. ECE1c protein levels were evaluated using Western blots with an anti-Flag antibody, using β-actin as a loading control. Representative blots from three independent experiments are shown. Representative Western blots of ECE1c^WT^ or ECE1c^K6R^ and β-actin from three independent cell lines (*n* = 3).

**Figure 3 cells-12-00506-f003:**
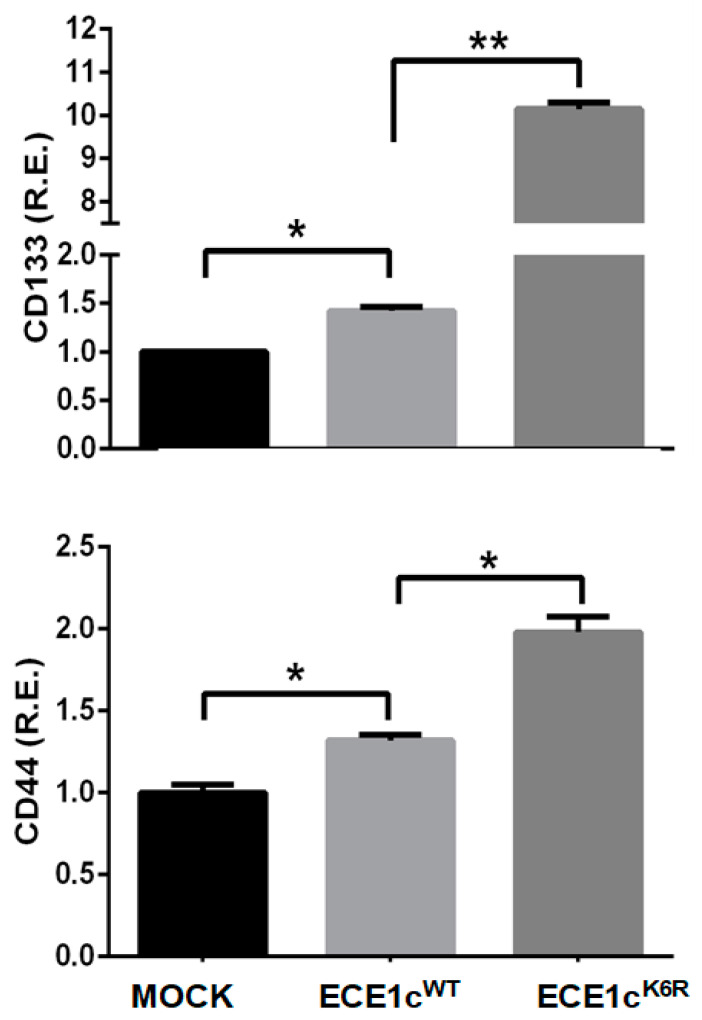
ECE1c^K6R^ expression promoted increases in CD133 and CD44 transcript levels. U87MG cells expressing either Flag-tagged ECE1c^WT^ or ECE1c^K6R^ proteins were grown under normal conditions for 24 h. Then, mRNA levels of CD133 and CD44 genes were quantified using RT-qPCR. Data represent averages ± SEMs (*n* = 3). Peritz F and Student’s tests were used. * *p* ≤ 0.05, ** *p* ≤ 0.01.

**Figure 4 cells-12-00506-f004:**
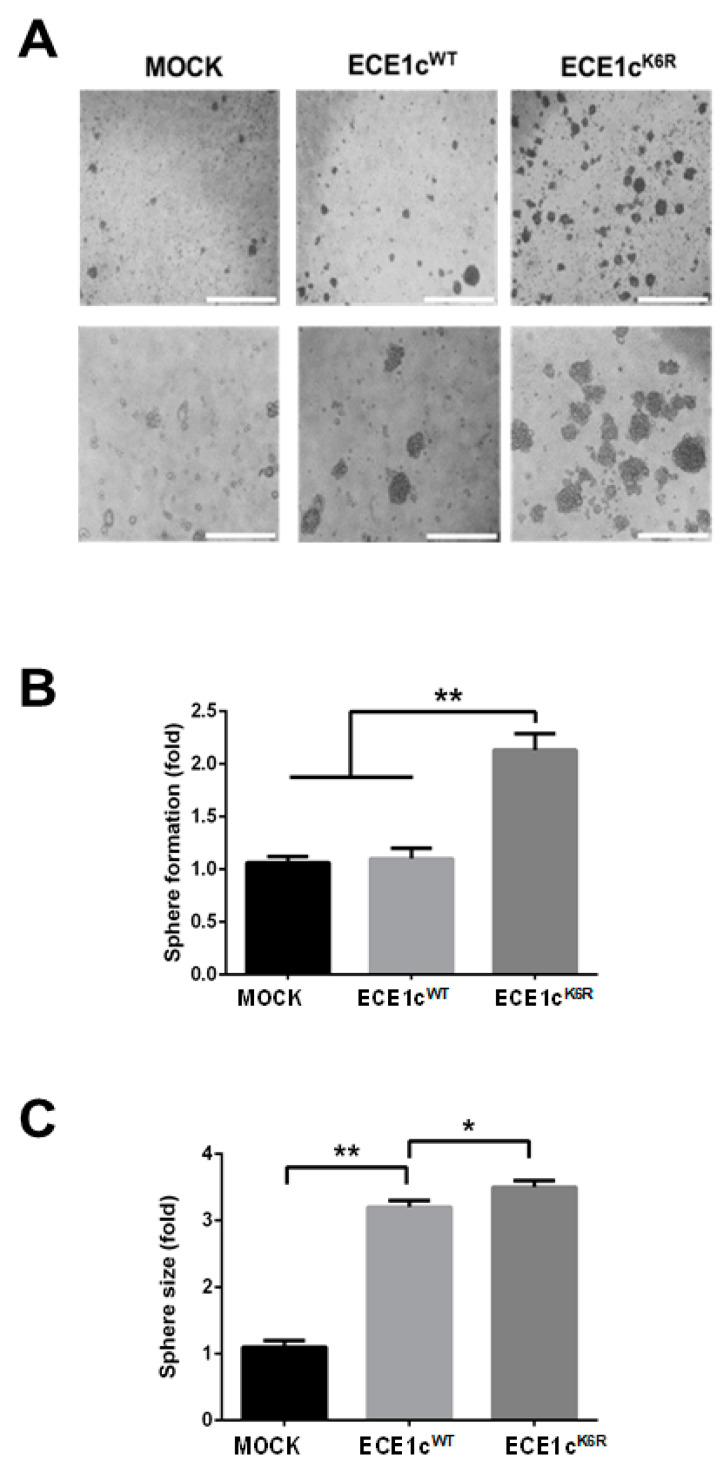
GSC enrichment was promoted by ECE1c^K6R^. (**A**) GSC enrichment was assessed using U87MG clone cells expressing ECE1c^WT^ or ECE1c^K6R^ and mock cells after 24 h of growth in a neurobasal medium for 7 days (i.e., a neurosphere-forming saturated condition, as detailed in the Materials and Methods). Representative images from three independent experiments are shown. (**B**) The number of spheres formed in A was determined from a triplicate analysis. (**C**) The sizes of the spheres formed in A were quantified from a triplicate analysis. Data represent averages ± SEMs (*n* = 3). Peritz F and Student’s tests were used. * *p* ≤ 0.05, ** *p* ≤ 0.01.

**Figure 5 cells-12-00506-f005:**
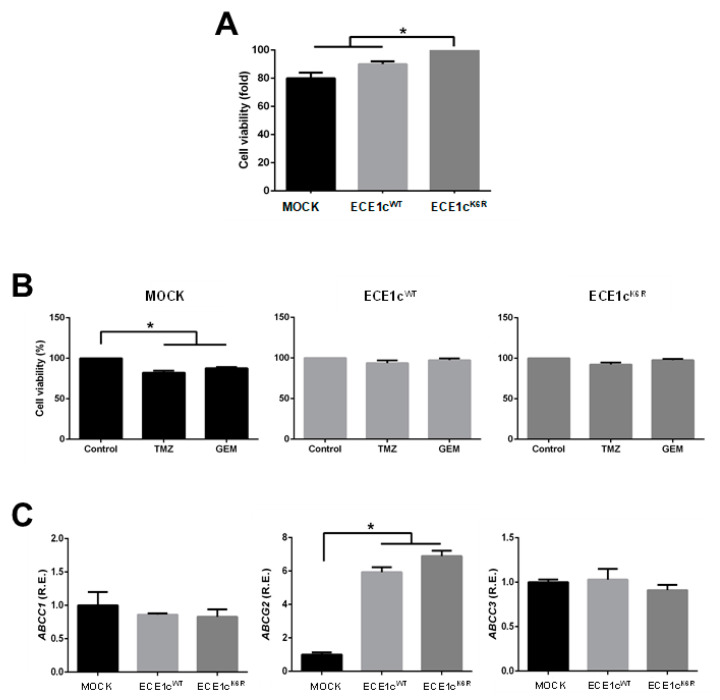
ECE1c overexpression conferred resistance to temozolomide and gemcitabine. (**A**) Basal viability of mock, ECE1c^WT^-expressing, and ECE1c^K6R^-expressing U87MG clone cells was measured after 24 h of growth in normal conditions using an MTS assay. (**B**) The chemoresistance of cells grown as in A was determined in the presence of either 400 μM temozolomide (TMZ) or 4 μM gemcitabine (GEM). DMSO was used as a vehicle (control). (**C**) Cells grown as in A under normal conditions were analyzed for the mRNA levels of the ABCC1, ABCG2, and ABCC3 genes using RT-qPCR. Data represent averages ± SEMs (*n* = 3). Peritz F and Student’s tests were used. * *p* ≤ 0.05.

**Figure 6 cells-12-00506-f006:**
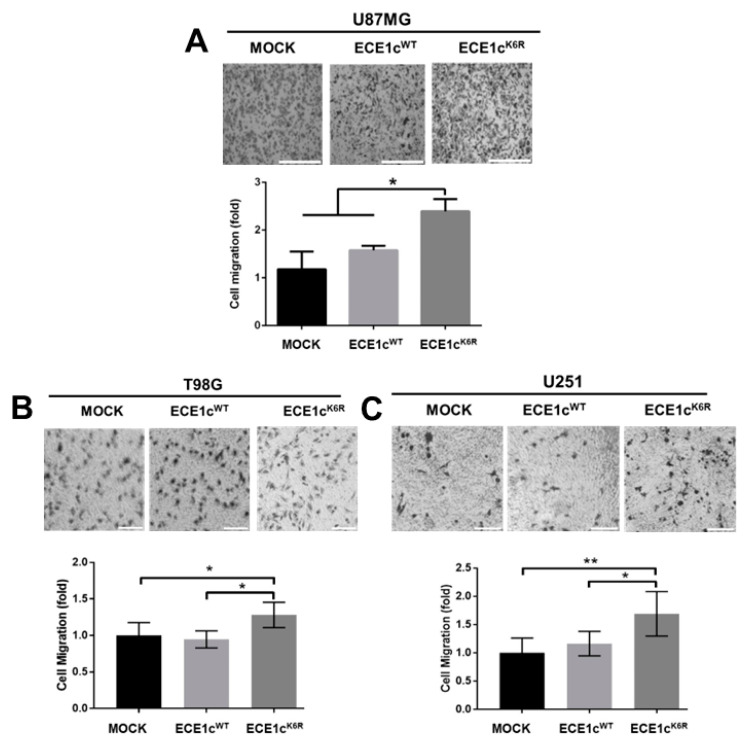
Migration was improved in ECE1c^K6R^-expressing GBM cells. Three-dimensional cell migration was evaluated in mock, ECE1c^WT^-expressing, and ECE1c^K6R^-expressing U87MG (**A**), T98G (**B**), and U251 (**C**) cells grown in normal conditions using a transwell assay, with 10% FBS as a chemoattractant. Cells were incubated at 37 °C for 2 h, and cell that migrated were fixed and stained with crystal violet. Cells were counted using the 4x objective in 5 different fields, mock normalized to 1, and plotted as fold changes. Scale bar: 100 nm. Data represent averages ± SEMs (*n* = 3). Peritz F and Student’s tests were used. * *p* ≤ 0.05, ** *p* ≤ 0.01.

**Figure 7 cells-12-00506-f007:**
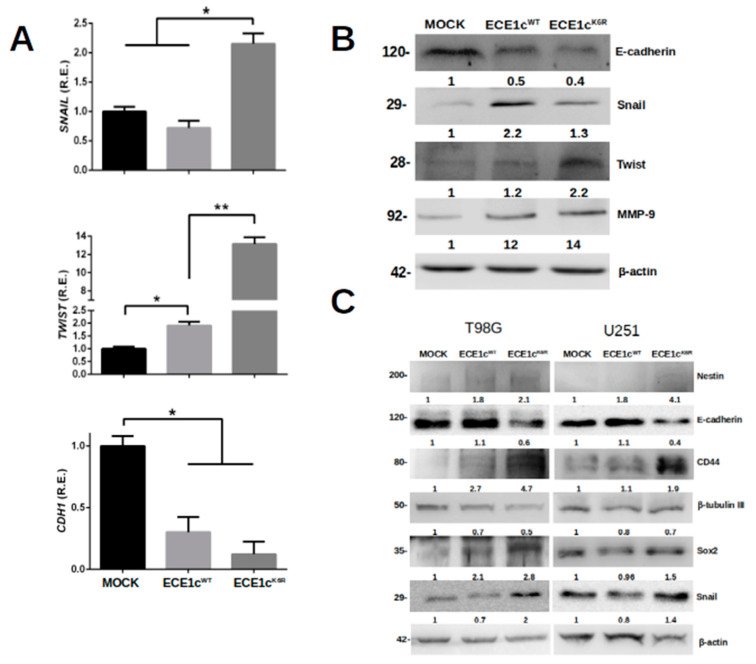
ECE1c^K6R^ overexpression promoted the expression of EMT-linked markers. (**A**) Transcript levels of the EMT markers Snail, Twist, and E-cadherin of mock, ECE1c^WT^-expressing, and ECE1c^K6R^-expressing U87MG clone cells grown in normal conditions for 24 h were quantified using RT-qPCR. Data represent averages ± SEMs (*n* = 3). Peritz F and Student’s tests were used. * *p* ≤ 0.05, ** *p* ≤ 0.01. (**B**) Protein levels of EMT-related markers of cells grown as in A were measured using Western blots with specific antibodies. (**C**) Protein levels of both stemness- and EMT-related markers of mock, ECE1c^WT^-expressing, and ECE1c^K6R^-expressing T98G or U251 clone cells grown in normal conditions for 24 h were measured using Western blots with specific antibodies. Representative images from three independent experiments are shown.

**Figure 8 cells-12-00506-f008:**
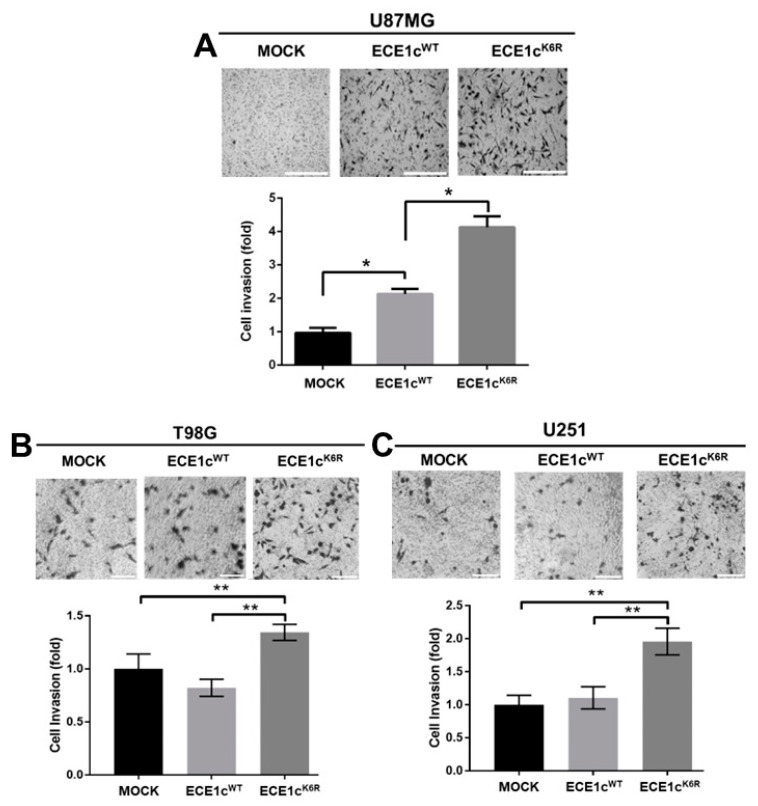
Cell invasion was improved in GBM cells expressing ECE1c^K6R^. Cell invasion was evaluated using a Matrigel-coated assay in mock, ECE1c^WT^-expressing, and ECE1c^K6R^-expressing U87MG (**A**), T98G (**B**), and U251 (**C**) clone cells grown in normal conditions for 6 h. Cells were grown in normal conditions at 37 °C for 6 h with 10% FBS as a chemoattractant. Then, the invaded cells were fixed and stained with crystal violet. Cells were counted using the 4X objective in 5 different fields, mock normalized to 1, and plotted as fold changes. Data represent averages ± SEMs (*n* = 3). Peritz F and Student’s tests were used. * *p* ≤ 0.05, ** *p* ≤ 0.01.

**Figure 9 cells-12-00506-f009:**
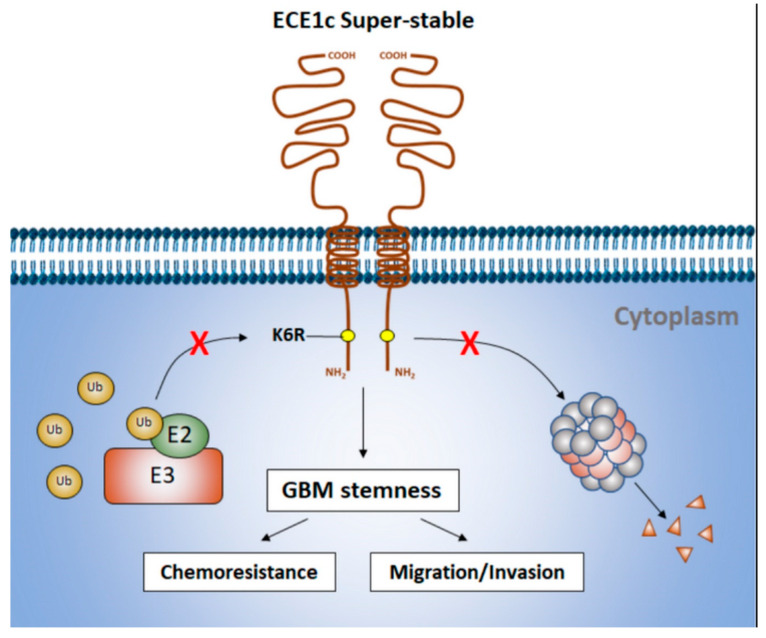
Proposed model. Lysine 6 (K6) is a plausible target for ubiquitination. Therefore, K6R mutation promotes a more stable ECE1c, thereby enhancing stem-like traits such as chemoresistance and cell migration/invasion. The mechanism of ubiquitination and the proteins involved in this process remain unclear and are discussed in Section 4.

## Data Availability

Not applicable.

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
