# Peer review of "Cancer Stem Cell and Aggressiveness Traits Are Promoted by Stable Endothelin-Converting Enzyme-1c in Glioblastoma Cells"

_cells, 2023, doi:10.3390/cells12030506_

Round 1

Reviewer 1 Report

Ref: Cells-2135302

The manuscript entitled: “Cancer stem cell and aggressiveness traits are promoted by stable endothelin-converting enzyme-1c in glioblastoma cells” by Niechi et al. is a research article that suggests phospho-ECE1c as a novel marker and potential therapeutic target in GBM. This is an interesting study and a well written manuscript, that fits within the scope of the journal. The manuscript needs improvement before acceptance for publication in the Cells journal after major revisions. Please find below the comments-suggestions and revisions that will help the authors improve the current version of this manuscript:

Major/minor comments:

Abstract:

-Lines 45-46: “such as elevated adhesion, migration, and invasion, which correlated with known EMT markers”: would be better to be rephrased to “such as modulation of adhesion, enhancement of cell migration and invasion”.

Methods:

-Line 125: “then suspended in 500 l 250 mM”: please correct this statement since something is missing here.

-Line 132: “Expression of mCherry was examined 72 h post-transduction”: the word "post-transfection" could be more suitable here than "post-transduction".

-Line 138:” Cells were grown at 37°C”: please define the type of the cells here.

-Line 145: For the cell viability more than one time point should have been tested (i.e 48h and 72h). Also the authors should justify the choice of only one concentration of the drugs tested and not a gradient first to test the potency of each drug and then a selection of 3-4 concentrations for the experiment would be more solid here.

-Line 147: Cell adhesion: please write the reference that this protocol was taken and/or the company`s kit. This is also a general comment for the methodology part-please address the methods obtained with more details.

-Line 156: IIF- spheres were seeded: please describe how these spheres were produced briefly.

-Line 173-174: Vascular mimicry: please define the source of the protocol and also explain the use of ovarian cells here instead of glioma cells.

Results:

-Figure 1 legend: please be careful the way you write this legend as some sentences (and numbers included) are a bit confused here-re-write with more clarity.

-Lines 273-274: “As CD44 and CD133 markers have been related to a stem-like phenotype and used for GSC detection (38–40), mRNA levels were quantified by RT-qPCR in U87MG cell clones”: please explain the use of the U87MG cell type and not others. Also, an IF experiment and/or western blot experiment to further validate the mRNA expression at the protein level for both CD44 and CD133 is essential here.

-Line 280: “ECE1cK6R promotes expression of stemness markers”: please rephrase this sentence to reflect better the meaning.

-Figure 4: be careful with the different panels to be all included here and in agreement with the legend.

-Figure 5C: mRNA levels (RT-qPCR results) should be further confirmed at the protein level (using western blot for instance).

-Figure 7B-C: Some of the western blot images in the main figure (supplementary) need to be significantly improved (for instance MMP9 in supplementary; 7C: CD44, COX2). Please check and improve the quality of the figures of these western blot experiments (perhaps by repeating some of these experiments).

-Line 356: “Firstly, ET1 extracellular levels was measured by ELISA.”: please describe briefly this methodology in the methods section.

-Supplementary images for western blot are not of good quality; please try to improve this.

Discussion:

-Lines 383-384: "Thus, ECE1c may contribute to aggressiveness in an epigenetic manner”: the authors may also mean the post-translational modification manner here since the phosphorylation is referred to the protein level (PTMs). Please elaborate.

-Lines 445-446: “suggesting that secreted ET1 is not enough to explain the enhanced aggressiveness observed in our GBM cell clones.”: please mention and explain what else could be done to be more conclusive over this finding (for instance which other experiments could be done etc).

- Line 460-461: “Thus, that a super-stable ECE1c leads to an aggressive GSC phenotype in GBM cells is an issue that may be occurring in patients”: please expand more and write more details of how you could associate your results with poor prognosis, in relation to the associated mRNA studies  (TCGA data) and your in vitro findings.

Author Response

Major/minor comments:

Abstract:

-Lines 45-46: “such as elevated adhesion, migration, and invasion, which correlated with known EMT markers”: would be better to be rephrased to “such as modulation of adhesion, enhancement of cell migration and invasion”

Changes were made according to the reviewer's suggestions and highlighted in yellow in the text.

Methods:

-Line 125: “then suspended in 500 l 250 mM”: please correct this statement since something is missing here.

Changes were made according to the reviewer's suggestions and highlighted in yellow in the text.

-Line 132: “Expression of mCherry was examined 72 h post-transduction”: the word "post-transfection" could be more suitable here than "post-transduction".

A lentivirus was used to infect the GBM cells, the word was changed to post-infection. The change is highlighted in yellow.

-Line 138:” Cells were grown at 37°C”: please define the type of the cells here.

Change was made according to the reviewer's suggestions and highlighted in yellow in the text.

-Line 145: For the cell viability more than one time point should have been tested (i.e 48h and 72h). Also the authors should justify the choice of only one concentration of the drugs tested and not a gradient first to test the potency of each drug and then a selection of 3-4 concentrations for the experiment would be more solid here.

In our laboratory we use this concentration and time since we previously standardized that at 24 h a chemosensitization effect is observable. The concentration was also previously determined for both drugs (data not shown in the paper). Considering the reviewer's suggestion, this explanation was added to the main text and highlighted in yellow.

-Line 147: Cell adhesion: please write the reference that this protocol was taken and/or the company`s kit. This is also a general comment for the methodology part-please address the methods obtained with more details.

We did not use a kit, but a standard protocol with common reagents was performed previously in our laboratory (the reference was added in the main text and highlighted in yellow). Considering the reviewer's suggestion, methods were explained with more detail.

-Line 156: IIF-spheres were seeded: please describe how these spheres were produced briefly.

Change was made according to the reviewer's suggestions and highlighted in yellow in the text. This procedure is explained in point 2.4 of M&M.

-Line 173-174: Vascular mimicry: please define the source of the protocol and also explain the use of ovarian cells here instead of glioma cells.

The VM protocol has been successfully used in reference 38 in which HEY-A8 were used to determine the optimal number of cells needed to get vascular structures. Thus, here we have taken conditioned media from either ECE1cwt- or ECE1cK6R-expressing T98 and U251 cells to study whether ET1 in these conditioned media may alter the capacity of HEYA8 cells to undergo VM. Considering the reviewer's suggestion, this explanation was added to the main text (M&M) and highlighted in yellow.

Results:

-Figure 1 legend: please be careful the way you write this legend as some sentences (and numbers included) are a bit confused here-re-write with more clarity.

Figure 1 legend was corrected according to the reviewer’s suggestion.

-Lines 273-274: “As CD44 and CD133 markers have been related to a stem-like phenotype and used for GSC detection (38–40), mRNA levels were quantified by RT-qPCR in U87MG cell clones”: please explain the use of the U87MG cell type and not others. Also, an IF experiment and/or western blot experiment to further validate the mRNA expression at the protein level for both CD44 and CD133 is essential here.

We added CD44 and Nestin IF in U87 MG cells (Fig 5S). We further assessed CD44 and Nestin levels by WB in U251 and T98G cells (Fig 7C).

-Line 280: “ECE1cK6R promotes expression of stemness markers”: please rephrase this sentence to reflect better the meaning.

Change was made according to the reviewer's suggestion to reflect a better meaning of our results and highlighted in yellow.

-Figure 4: be careful with the different panels to be all included here and in agreement with the legend.

Change was made in figure legend according to the reviewer's suggestion and highlighted in yellow.

-Figure 5C: mRNA levels (RT-qPCR results) should be further confirmed at the protein level (using western blot for instance).

We appreciate the reviewer's suggestion and agree that it is necessary to confirm protein levels by western blot. However, currently we do not have antibodies against the MRP1, BCRP, and MRP3 proteins. Thus, we adjusted our conclusions in accordance with the RT-qPCR results on the understanding that we evaluated only this but not protein levels.

-Figure 7B-C: Some of the western blot images in the main figure (supplementary) need to be significantly improved (for instance MMP9 in supplementary; 7C: CD44, COX2). Please check and improve the quality of the figures of these western blot experiments (perhaps by repeating some of these experiments).

Changes were made in figure7B and 7C according to the reviewer's suggestion.

-Line 356: “Firstly, ET1 extracellular levels was measured by ELISA.”: please describe briefly this methodology in the methods section.

Change was included in the M&M Section 2.13. Enzyme-Linked Immunosorbent Assay (ELISA) highlighted in yellow.

-Supplementary images for western blot are not of good quality; please try to improve this.

The images were improved and added in TIFF format.

Discussion:

-Lines 383-384: "Thus, ECE1c may contribute to aggressiveness in an epigenetic manner”: the authors may also mean the post-translational modification manner here since the phosphorylation is referred to the protein level (PTMs). Please elaborate.

Change was made according to the reviewer's suggestion and highlighted in yellow.

-Lines 445-446: “suggesting that secreted ET1 is not enough to explain the enhanced aggressiveness observed in our GBM cell clones.”: please mention and explain what else could be done to be more conclusive over this finding (for instance which other experiments could be done etc).

As stated in M&M 2.13 (ELISA), ET1 levels were measured after 48 h of growth, which may be too late to observe significant differences between ECE1cWT and ECE1cK6R clones. Therefore, differential ET1 levels during earlier stages cannot be ruled out, with a potentially significant impact on the stemness traits observed during later stages of growth. This has been included at Discussion of the revised version and highlighted in yellow.

- Line 460-461: “Thus, that a super-stable ECE1c leads to an aggressive GSC phenotype in GBM cells is an issue that may be occurring in patients”: please expand more and write more details of how you could associate your results with poor prognosis, in relation to the associated mRNA studies (TCGA data) and your in vitro findings.

We think our data uncover the existence of an post-translational mechanism enhancing aggressiveness of GBM cells, which involves the CK2-dependent phosphorylation of ECE1c, as no mutations at K6, S18, or S20 have been found (personal communication). Our findings suggest that in GBM, ECE1c (augmented according to TCGA) may be phosphorylated at S18 and S20 by CK2 (also augmented as shown at 23 and 24 promoting stemness traits and becoming the tumor more aggressive, leading to recurrence and/or metastasis and thereby poor patient prognosis. Thus, ECE1c could contribute to cancer aggressiveness in an epigenetic manner as we have suggested elsewhere [20]. This has been included at Discussion of the revised version and highlighted in yellow.

Reviewer 2 Report

Reviewer (Remarks to the Author):

The manuscript entitled “Cancer stem cell and aggressiveness traits are promoted by stable endothelin-converting enzyme-1c in glioblastoma cells.” by Ignacio Niechi et al. suggestes tphospho-ECE1c as a novel marker for poor prognosis, as well as a potential therapeutic target for GBM. The premise of the work is very interesting, however in its present version, the manuscript requires several significant areas of improvement before consideration for publication.

1) Overall the quality of the microscope data (Figure 4A, 6A, 6B, 6C, 8B, 8C) are not particularly strong, often hard to understand. The scale bar information for images should be added to images and Figure Legends, making it difficult for the reader to assess the size between different images.

2) How many independent experiments were done for the western blots. For the western blot loading control protein seems very strong (Figure 2C, 3C). There is a good chance that the normalization may be done incorrectly? It is better to make border for the western images and add analysis for all western blots?

3) Were different gels used for these proteins for Figure 7B and 7C western image? The molecular weight of Nestin or E-cadherin is around 150 kDa. The SnaII and Twist bands are very close to each other? How did you separate the bands? Did you run different gels or strip it? If you use different gels, it is better show for all proteins individual beta actins?

4) Were different gels used for these proteins for Figure 7C? The molecular weight of  Sox2 is around 37 kDa. The band of Sox2 is very close to molecular weight of beta actin? How did you separate the bands? Did you run different gels or strip it? If you use different gels, it is better to show for all proteins individual beta actins?

5) It would be more interesting if authors can provide one separate model Figure showing the results of manuscript.

6) The statistical approach is not sufficiently described and explained.  In particular, the number of experimental replications is not mentioned in the text or figure legends.  The nature of variability in "normalized" controls is not explained.  The nature of corrections for multiple comparisons is not provided.

7) The descriptions of data in the figures also needs to be significantly improved.

8) Although the manuscript shows some interesting correlative data, there are some issues with quality of Figures. The quality of Figures needs to improved.

9) If cell lines were used in the research, a statement addressing the following points must be included in the Materials and Methods section of the manuscript.

b. Whether the cell lines have been tested and authenticated

 c. The method by which the cells were tested

 d. How and when the cells were last tested

If cells were obtained directly from a cell bank that performs cell line characterizations and passaged in the user's laboratory for fewer than 6 months after receipt or resuscitation, re-authorization is not required. In these cases, please include the method of characterization used by the cell bank.

10) The text needs careful proof reading.

11) Line 163, Line 148: Cells (7.5×104 cells/chamber) were plated on. 104 has to be corrected.

Author Response

1) Overall the quality of the microscope data (Figure 4A, 6A, 6B, 6C, 8B, 8C) are not particularly strong, often hard to understand. The scale bar information for images should be added to images and Figure Legends, making it difficult for the reader to assess the size between different images.

The images were improved and added in tiff format. Scale bar was added and legends were improved.

2) How many independent experiments were done for the western blots. For the western blot loading control protein seems very strong (Figure 2C, 3C). There is a good chance that the normalization may be done incorrectly? It is better to make border for the western images and add analysis for all western blots?

Three independent experiments were performed and the densitometric analysis for western blots was included (numbers below each membrane normalized with beta-actin)

3) Were different gels used for these proteins for Figure 7B and 7C western image? The molecular weight of Nestin or E-cadherin is around 150 kDa. The SnaII and Twist bands are very close to each other? How did you separate the bands? Did you run different gels or strip it? If you use different gels, it is better show for all proteins individual beta actins?

We corrected the molecular weight of each protein, since we previously only showed the molecular weight closest to the ladder. Stripping was performed in cases of close MW similarity, like Snail and Twist.

4) Were different gels used for these proteins for Figure 7C? The molecular weight of Sox2 is around 37 kDa. The band of Sox2 is very close to molecular weight of beta actin? How did you separate the bands? Did you run different gels or strip it? If you use different gels, it is better to show for all proteins individual beta actins?

We performed stripping to blot proteins with similiar molecular weight.

5) It would be more interesting if authors can provide one separate model Figure showing the results of manuscript.

According to the reviewer's suggestion a model was added (Figure 9).

6) The statistical approach is not sufficiently described and explained.  In particular, the number of experimental replications is not mentioned in the text or figure legends.  The nature of variability in "normalized" controls is not explained.  The nature of corrections for multiple comparisons is not provided.

Experiments were performed at least three times (n=3). The "n" and the statistical tests used for each analysis were included in the legend of each figure and highlighted in yellow.

7) The descriptions of data in the figures also needs to be significantly improved.

Figure legends have been improved as requested.

8) Although the manuscript shows some interesting correlative data, there are some issues with quality of Figures. The quality of Figures needs to improved.

Figures quality have been improved as requested.

9) If cell lines were used in the research, a statement addressing the following points must be included in the Materials and Methods section of the manuscript.

b. Whether the cell lines have been tested and authenticated

 c. The method by which the cells were tested

 d. How and when the cells were last tested

If cells were obtained directly from a cell bank that performs cell line characterizations and passaged in the user's laboratory for fewer than 6 months after receipt or resuscitation, re-authorization is not required. In these cases, please include the method of characterization used by the cell bank.

All cell lines were purchased from ATCC (Manassas, VA). U87MG cells, the main line used in this work, were purchased by Dr. Quezada and the invoice is attached together this letter. T98G and U251 cells were purchased and gently donated by Dr. Varas-Godoy. Once arrived at laboratory, cells were immediately expanded in medium supplemented with 10% FBS, 100 U/ml penicillin and 100 μg/ml streptomycin (Gibco) at 37°C and 5% CO2, followed by storage in liquid nitrogen at -190°C. Once a year, one N2 aliquot was thawed, expanded, and stored again at -80°C. For experiments, one -80°C aliquot was thawed and grown in normal media. All experiments were performed within one year and cells eliminated after 15 passages, as requested by each local biosecurity committee. Mycoplasma contamination was tested monthly with the EZ-PCR Mycoplasma Test kit (Biological Industries, Beit Haemek, Israel), being the last test performed six months ago and yielding no contamination. This has been included at M&M (2.1 section) of the revised version and highlighted in yellow.

10) The text needs careful proof reading.

Main text have been improved as requested.

11) Line 163, Line 148: Cells (7.5×104 cells/chamber) were plated on. 104 has to be corrected.

Change was made according to the reviewer's suggestion and highlighted in yellow.

Round 2

Reviewer 1 Report

Ref: Cells-2135302

The manuscript entitled: “Cancer stem cell and aggressiveness traits are promoted by stable endothelin-converting enzyme-1c in glioblastoma cells” by Niechi et al. is an interesting research article. The manuscript has now improved therefore I believe can be considered for publication in the Cells journal  after 1 minor correction:

-lines 236-237: ELISA results should have been expressed as ET-1 concentration (ng/ml) normalized to 1 mg/ml of total protein.

Author Response

The change was made according to the reviewer's suggestions and was highlighted in yellow. As the reviewer indicates,  the correct form is "expressed as ET-1 concentration (pg/ml) normalized to 1 mg/ml of total protein".